# Fructans as Immunomodulatory and Antiviral Agents: The Case of *Echinacea*

**DOI:** 10.3390/biom9100615

**Published:** 2019-10-16

**Authors:** Erin Dobrange, Darin Peshev, Bianke Loedolff, Wim Van den Ende

**Affiliations:** 1Laboratory of Molecular Plant Biology, KU Leuven, 3001 Leuven, Belgium; erin.dobrange@kuleuven.be (E.D.); darinpeshev@gmail.com (D.P.); 2Institute for Plant Biotechnology, Department of Genetics, Faculty of AgriSciences, Stellenbosch University, Matieland 7602, South Africa; bianke@sun.ac.za

**Keywords:** antioxidants, *Echinacea*, fructans, immunomodulation, prebiotics, herbal medicine

## Abstract

Throughout history, medicinal purposes of plants have been studied, documented, and acknowledged as an integral part of human healthcare systems. The development of modern medicine still relies largely on this historical knowledge of the use and preparation of plants and their extracts. Further research into the human microbiome highlights the interaction between immunomodulatory responses and plant-derived, prebiotic compounds. One such group of compounds includes the inulin-type fructans (ITFs), which may also act as signaling molecules and antioxidants. These multifunctional compounds occur in a small proportion of plants, many of which have recognized medicinal properties. *Echinacea* is a well-known medicinal plant and products derived from it are sold globally for its cold- and flu-preventative and general health-promoting properties. Despite the well-documented phytochemical profile of *Echinacea* plants and products, little research has looked into the possible role of ITFs in these products. This review aims to highlight the occurrence of ITFs in *Echinacea* derived formulations and the potential role they play in immunomodulation.

## 1. Herbal Medicines Comprise the Second Largest Therapeutic Market Globally

The use of herbal medicine is an ancient practice that relies on the use of a broad range of plant and plant-derived products (botanical materials) for medicinal purposes. These herbal products are prepared and used in various forms, including capsules (dry powder), tinctures (ethanol extracts), infusions (extractions usually based on water or oil, such as tea), macerations (soaking of plant material), and decoctions (boiling of plant material). The largest proportion (80%) of communities in developing countries (particularly Asia and Africa) relies on herbal medicine and associated herbal preparations as primary healthcare. Compared to other therapeutics used in modern medicine (MM), herbal medicines are perceived as being more affordable, accessible, and acceptable to the communities using it [1,2]. The use of herbal products as an alternative and complementary to MM created a global market recognized by the World Health Organization (WHO) as the second-largest therapeutics market worldwide [3,4]. The uncertainties (pertaining to efficacy and safety) arising around the use of MM therapeutics is one of the driving factors to search for alternative or complementary medicines in the form of natural resources. In this regard, the use of herbal medicines for immunomodulation could, therefore, provide an effective and safe complement to MM [5,6,7]. Consequently, research efforts largely focus on identifying and investigating specific groups of plant-related compounds (such as flavonoids, polysaccharides, lactones, alkaloids, diterpenoids, and glycosides) and their potential implication in immunomodulation [8,9,10]. Typically, in crude extracts, these compounds occur together in a mixture. Understanding the relative contribution of each compound to the overall immunomodulatory effect is usually a challenging task.

Immunomodulators could act (stimulate or suppress) on both the innate- and adaptive immune systems to (i) exert positive effects on the host defense mechanisms, (ii) have an anti-proliferative effect on tumor cells and, (iii) enhance the host’s ability to tolerate damages caused by toxic compounds (such as chemotherapeutics). *Echinacea* preparations are among the most popular herbal products with immunomodulatory properties [11,12]. While immunomodulatory aspects of an array of different biomolecules (e.g., phenolic compounds, alkamides, arabinogalactan proteins) occurring in *Echinacea* preparations have been highlighted before [13,14,15,16,17], here we review knowledge on *Echinacea* derived inulin-type fructans (ITFs) and highlight their presence in commercially available *Echinacea* products. The potential contribution of ITFs to immunomodulatory effects in the human body is discussed.

## 2. *Echinacea* Plant Extracts Exert Immunomodulatory Properties

*Echinacea* is a plant genus within the family Asteraceae (previously termed the Compositae) and is comprised of 11 taxa of herbaceous and flowering plants [18,19]. *Echinacea* preparations (which are mainly based on three commercially important species; *Echinacea purpurea*, *Echinacea angustifolia*, and *Echinacea pallida*) are commonly used for preventing and alleviating the symptoms of bacterial and viral infections [11,12,18]. Furthermore, some *Echinacea* preparations are known to exert antioxidant and anti-inflammatory activity espousing to its potential immunomodulatory activities [12,19,20].

Both *in vitro* and *in vivo* studies demonstrated that different extracts from *Echinacea*, either from roots, above-ground parts, or a mixture of both, could stimulate and signal immune responses [21]. Both aqueous extracts and alcoholic extracts are used. Differences between the species, the soil in which the plants are cultivated, the parts of the plant used, and the extraction procedures can result in substantial differences in chemical composition and biological activity [21]. Immunological studies have used aqueous or alcoholic extracts, as well as purified polysaccharides. Their effects on monocytes, macrophages, natural killer (NK) cells, T cells, and dendritic cells (DC) have been thoroughly studied [20,21,22]. A number of these immune-related cells are equipped with Pattern Recognition Receptors (PRRs), which are specialized in recognizing Microbe Associated Molecular Patterns (MAMPs) and Damage Associated Molecular Patterns (DAMPs). Collectively, these factors regulate the homeostasis of the immune system to maintain a state of health in the host. Further research concluded that several purified compounds from *Echinacea* (e.g., glycoproteins, soluble polysaccharides, caffeic acid derivatives, phenolic compounds and alkamides) could induce transcriptional changes that activate immunomodulation pathways [12,19,23,24,25,26,27,28,29]. While the immunomodulatory properties for the polyphenol and alkylamide fractions present in *Echinacea* are well-described, relatively little information is available about the polysaccharide fraction. In general, immunomodulatory and prebiotic polysaccharides have been implicated in the overall health and well-being of humans by (i) enhancing physiological parameters (e.g., blood pressure, hematological parameters), (ii) increasing tolerance against pathogens and, (iii) modulating immune responses [30,31]. Of interest in this review is a group of fructose-based oligo- and polysaccharides collectively termed fructans.

## 3. General Function of Fructans in the Human Body

Fructans are water-soluble, sucrose-derived compounds that occur in about 15% of flowering plants [32], including the economically important Poaceae and Asteraceae (including *Echinacea*) families. Based on structural differences, three types of fructans can be distinguished: inulin (β2→1 linkage), levan (β2→6 linkage), and graminan (β2→1 and β2→6 linkages). The chain-length of plant fructans (defined by the degree of polymerization: DP) is variable, with a few species (e.g., *Viguiera discolor*; *Echinops ritro*) standing out, producing higher DPs by specific fructosyltransferases [33]. Short-chain fructooligosaccharides with a DP < 10 are often referred to as FOS, whereas long-chain polymers with DP > 10 are usually called inulin or ITFs [34], well-known prebiotics contributing to overall health and well-being. Short-chain FOS are primarily metabolized in the proximal colon, while long-chain ITFs are fermented by bacteria in the distal colon, where many chronic diseases originate [35,36,37,38,39]. The beneficial immunomodulatory effects of fructans in the human gastrointestinal tract, and by extension to all other eukaryotic organisms, including animals and plants, are often attributed to their capacity to (i) stimulate the growth and proliferation of beneficial intestinal bacteria (prebiotics), (ii) act as signaling molecules and (iii) act as reactive oxygen species (ROS) scavengers [35,38,39,40,41,42]. These effects are, to a large extent, dependent on fructan DP [35,43].

The immunomodulatory function of ITFs was shown to be dependent on Toll-Like Receptors (TLRs), in particular, TLR2 and to a lesser extent, TLR4, 5, 7, 8 [35]. Fructans may be mimicking MAMPs. After binding to TLRs and used at lower doses, they may slightly trigger signaling pathways dependent on the nuclear factor NF-κB, resembling a ‘priming’ or ‘vaccination’ like effect [38]. However, an additional signaling pathway dependent on the peptidoglycan recognition protein 3 (PGlyRP3) has also been proposed [44]. This recognition protein, which responds to bacterial cell wall peptidoglycan, forms part of a larger group of highly conserved host defense proteins in mammals and insects. It has been demonstrated that FOS enhances the activity of peroxisome proliferator-activated receptors (PPAR), a group of nuclear receptor proteins that function as transcription factors to modulate the expression of several genes involved in lipid and carbohydrate metabolism, as well as cell proliferation, differentiation, and death [45,46]. The activation of peroxisome proliferator-activated receptor gamma (PPARγ) was shown to be induced by PGlyRP3, resulting in a decreased production of proinflammatory cytokines and inhibition of NF-κB-dependent signaling [44]. Further insights into the mechanism and modulation of these pathways would provide valuable information on the extent to which fructans contribute to immunomodulation.

### 3.1. Fructans as Immunomodulatory and Antiviral Compounds

Respiratory infections, such as the common cold and the seasonal flu, are often infectious diseases caused by different viruses [47]. Although vaccination is considered as an effective method to prevent infectious diseases, vaccination of healthy adults would only reduce the incidence of acute respiratory infections by 16% [5]. Apart from directly affecting an individual’s quality of life, the common cold and flu indirectly affects a society’s economy due to work absenteeism [26]. In turn, this is a driving factor for discovering alternative herbal medicines for the treatment of viral infections. In this regard, maintaining healthy homeostasis of the immune defense system (often by stimulating NK cell activity and salivary immunoglobulin A production) could play an important role in preventing viral infections. In this context, *Echinacea* extracts show profound antiviral activity against several viruses (including human and avian influenza viruses, H3N2-type IV, H1N1-type IV, herpes simplex, and rhinoviruses), and reversed virus-induced pro-inflammatory responses [48,49,50]. It is necessary to better understand what role ITFs and other fructans may play in immunomodulation, when administered as part of a collective plant extract, given the knowledge that a variety of other herbal medicines (such as the Indian Ayurveda and Japanese Chikuyo–Sekko–To) with well-established immunomodulatory effects have been identified to contain fructans as key compounds [51,52].

Fructans have been demonstrated to exert immunomodulatory and antiviral effects directly. An ITF (CSH1-1, Mw = 4.0 × 10^3^) isolated from traditional Japanese herbal medicine (Chikuyo–Sekko–To) was shown to be effective against herpes simplex virus type 2 (HSV-2) in a murine model [51]. The antiviral properties were attributed to the ability of the fructan to enhance the production of nitric oxide (NO; a viral replication inhibitor) and other immunostimulatory factors (e.g., interleukine-1 beta (IL)-1β, IL-6, IL-10, interferon gamma (IFN)-γ, and tumor necrosis factor alpha (TNF)-α) on RAW264.7 cells. Similarly, ITFs from chicory (*Cichorium intybus*) are known to increase NO production in INF-γ-primed RAW 264.7 cells in an NF-κB-dependent manner [53]. A fructan isolated from Welsh onion (*Allium fistulosum L.*) demonstrated an inhibitory effect on influenza A virus replication in mice [54]. Intriguingly, fructans from burdock (*Arctium lappa L.*) strongly stimulate NO synthesis and defense signaling in plants [55], suggesting that the overall underlying mechanisms and pathways may be similar in all multicellular organisms. Both high and low DP fructans from aged and fresh garlic (*Allium sativum*) have the capacity to activate macrophages and subsequently phagocytosis, again in combination with a release of NO [56,57]. Garlic is also an important component of the traditional Ayurveda Rasayana drugs, together with the *Inula racemosa* and *Bombax ceiba*. The immunomodulating properties of these three plants were attributed to their high fructan content [58,59,60]. Fructans were also shown to be important immunomodulatory compounds in extracts of onion (*Allium cepa*), yacon (*Smallanthus sonchifolius*), Curcuma (*Curcuma kwangsiensis*), blue agave (*Agave tequilana*), and mugwort (*Artemisia vulgaris*) [61,62,63,64,65].

### 3.2. Fructans as Antioxidative and Anti-inflammatory Compounds

Fructans and other plant-derived sugars are known to exert antioxidant properties by directly and indirectly affecting ROS balances and countering the damage caused by oxidative stress [41,66,67,68]. Oxidative stress is further known to trigger inflammation (an innate immune response) and, although an important response in the human body, it has been linked to severe autoimmune disorders. *Echinacea* and several other plants encompassed in herbal medicine (e.g., ginger, turmeric, and cannabis) have been shown to alleviate the effects of inflammation. It is interesting to note that these plants display enhanced antioxidant capacity, which could, in turn, be used to manage inflammation-related disorders induced by oxidative stress [69].

Fructan-related antioxidant capacity (and ability to act as ROS scavengers) has also been demonstrated in humans in both *in vitro* and *in vivo* studies, suggesting that the consumption of foods rich in fructans could exert beneficial immunomodulatory properties by acting as antioxidants [68,70,71]. The ability of ITFs to bind to TLR2 suggests a potential role in anti-inflammatory pathways through the activation of regulatory T-cells (Tregs) [38,72]. In line with this reasoning, inulin was shown to create an immunosuppressive environment in human peripheral blood mononuclear cells (PBMC) by promoting the expression of forkhead box P3 (FOXP3; a Treg biomarker) and the secretion of the anti-inflammatory cytokine IL-10 [73]. TLR2 signaling is also known to decrease intestinal permeability, a condition directly linked to inflammatory intestinal diseases such as celiac disease, inflammatory bowel disease (IBD), and irritable bowel syndrome (IBS) [74,75]. Interestingly, it was demonstrated that ITFs, independently of their action on microbiota, have the ability to enhance the expression of fucosyltransferase 2 (FUT2), an enzyme responsible for the fucosylation of gut epithelial cells and considered to play a major role in immunomodulation [76]. ITFs may also have anti-inflammatory effects through the inhibition of NF-κB, which has been proposed by Zenhom et al. [44].

It is well-known that ITFs indirectly influence AMP-activated protein kinase (AMPK) signaling through modulation of the microbiota, for instance, by stimulating the growth of Lactobacilli and Bifidobacteria, producing short-chain fatty acids (SCFAs) that activate AMPK [77]. It is also possible that ITFs exert a more direct effect on AMPK, as previously proposed [40]. Metformin, a popular antidiabetic drug, is known to activate AMPK and to inhibit NF-κB activation [78]. Surprisingly, ITFs and metformin have very similar physical outcomes [79]. PPARγ, modulated by ITFs, is also known to influence AMPK signaling [80]. These properties of ITFs make them good candidates for the treatment of current inflammatory diseases such as diabetes, obesity and IBD [81,82]. The immunomodulatory and anti-inflammatory properties of *Echinacea* preparations are well-known and have been ascribed to the myriad of compounds that display antioxidant activity [83]. Both *Echinacea angustifolia* and *Echinacea purpurea* were shown to improve symptoms of colitis in rats by decreasing the number of inflammatory mediators (IL-1β and TNF-α) [84]. *Echinacea* preparations were also reported to reduce inflammatory conditions caused by insulin resistance and induced by pathogens, respectively [79,85,86,87,88,89].

## 4. ITFs and Polyphenols in *Echinacea* Preparations May Exert Synergistic Effects

Chicoric acid is a well-known antioxidant [68], and one of the major phenolic constitutes in *Echinacea* preparations, but its absorption from the gastrointestinal tract into the human body is not clear. Nevertheless, similar to what has been proposed for fructans, immunomodulatory, and anti-inflammatory effects, linked to AMPK signaling and downregulation of NF-κB, have also been suggested [90,91]. Moreover, Phuwamongkolwiwat et al. [92] suggested that FOS stimulated the uptake of phenolic compounds, an example of putative synergism between the two classes of biomolecules. Such synergistic effects between ITFs and polyphenols, although depending on entirely different mechanisms, were also suggested in chicory plant vacuoles [68,93,94]. Conversely, ITFs have been shown to reduce the bioavailability of phenolic compounds and subsequently, the total antioxidant function *in vitro*, suggesting that the interaction between phenolic compounds and ITFs is complex, particularly in the human system, and remain to be fully elucidated [95]. The speculation that phenolics, ITFs, other types of polysaccharides, and other compounds (e.g., alkamides through PPARγ signaling) may act together in a synergistic way is consistent with the widely accepted hypothesis that *Echinacea* preparations have such powerful immunomodulatory properties because of their complex composition [96].

## 5. Possible Significance of ITFs in Commercial *Echinacea* Preparations

*Echinacea* preparations are among the best-selling herbal medicines in the Western world for the treatment and possible prevention of upper respiratory infections, including the common cold and seasonal flu [11,12,26]. The mechanisms of action of such *Echinacea* preparations have usually been explained by a combination of antiviral, immunomodulating, anti-inflammatory, and antioxidant activities [12,25]. *Echinacea* preparations were reported to stimulate increases in both the number of white and red blood cells [97]. In line with the possible anticancer activities of *Echinacea* preparations [98], it has been demonstrated that *Echinacea* extracts are potent activators of NK cell cytotoxicity. *Echinacea* extracts increase the frequency of NK cell target conjugates and speed up their lytic activities [99]. It has been speculated that the action of *Echinacea* preparations is based on differential multi-level modulation of the responses of different types of leukocytes and T cells, regulating the production of chemokines and cytokines such as IL1-β, IL-6, IL-8, IL-10, IL-12, IFN-γ, and TNF-α [100,101,102,103]. Additionally, the modulation of DC activity, important for Treg induction, has been reported [20,22,104]. The polysaccharide fraction of *Echinacea* was shown to inhibit inflammatory responses by inhibiting the expression of the NF-κB p65 protein [105]. Furthermore, Echinaforce^®^ has been shown to prevent virus-induced bacterial adhesion by suppressing the expression of NF-κB [106]. There is a striking similarity between the above-discussed immunomodulatory properties of fructans and those of the *Echinacea* preparations.

Further research is necessary to decipher whether, and to what extent, ITFs contribute to the observed immunomodulatory effects of *Echinacea* preparations. The connection between the *Echinacea* polysaccharides and their immunomodulating properties has been suggested multiple times [23,24,107,108,109,110]. However, within the polysaccharide fraction, most research focused on the immune-stimulatory activity of cell-wall-derived arabinogalactans. Macrophage activation by the polysaccharide arabinogalactan isolated from plant cell cultures of *Echinacea purpurea* was already reported decades ago [111], but unfortunately, the inulin fraction in these polysaccharide preparations was completely neglected.

The occurrence of ITFs in *Echinacea* [112] prompted us to investigate whether fructans were present in an array of commercial *Echinacea* preparations. A range of low and high DP ITFs was found [113]. The average DP of the fructans present in the different preparations seem to be dependent on the method of extraction, with a higher occurrence of high DP fructans in dry extracts compared to ethanol extracts. The parts of the plant used were, however, not indicative of the relative fructan composition as both low and high DP fructans were found in preparations based on either above-ground or root material. The occurrence of fructans in *Echinacea* preparations was already reported by Giger et al. (1989), who found that the fructan content differed between *Echinacea angustifolia* and *Echinacea purpurea*, depending on the harvesting period [114]. They also found a tenfold lower fructan content in preparations based on the above-ground material compared to preparations based on root material. More recent studies also highlighted the presence of fructans in *Echinacea* preparations and found that the amount of ethanol used for extraction determines the fructan content of the extract [16,115]. As higher DP fructans are not soluble in ethanol, higher ethanol concentrations prevent the extraction of higher DP fructans while favoring a lower average DP.

We further demonstrated the total antioxidant capacity (TAC) and consequent DNA protection ability of the commercial products and the polysaccharide fractions, derived thereof, respectively. Not surprisingly, the products displayed significant TAC, as well as the ability to protect human DNA *in vitro*. Interestingly, we found that the polysaccharide fraction in two out the five products displayed significant differential TAC profiles, although not displaying clear DNA protection ability. In contrast, the polysaccharide fractions of the remaining three products displayed protective DNA abilities. We speculate that the occurrence of high DP fructans within polysaccharide fractions of these products could play some role in the efficacy thereof. Since ITFs only represent a small part of these preparations, it seems unlikely that prebiotic effects on their own would explain the observed immunomodulatory and health-promoting effects. More likely, ITFs present in *Echinacea* preparations stimulate the activity of immune cells directly by activating signaling pathways *in vivo*, as described [116,117,118]. Furthermore, it is important to note that the amount of ITFs required for prebiotic effects (8 to 12 g a day) is much higher than the recommended daily intake of a commercially available *Echinacea* preparation [8,119,120,121]. Taken together, the emerging evidence of the immunomodulating properties of ITFs in the literature and the present confirmation that commercial *Echinacea* products contain these fructans, strongly suggests that they could be among the contributors for the overall effect of these products.

## 6. Final Remarks

Numerous herb- and medicinal plant extracts have been successfully employed as immunomodulatory substances for maintaining human health. The resources of these extracts have historically been part of cultural traditions (such as Indian Ayurveda and Traditional Chinese Medicine) and have been carefully developed through anecdotal or pragmatic experience. Currently, a holistic approach based on a combination of modern therapeutics and herbal medicines is targeted for disease prevention. Various plant extracts are known to contain an array of active compounds with the ability to activate or regulate the immune system through direct or indirect mechanisms (such as signaling, stimulation, and modulation of macrophages, lymphocytes, and cytokine production). Immunomodulatory compounds from plant extracts include (i) flavonoids, (ii) glycosides, (iii) polysaccharides, (iv) terpenoids, (v) essential oils, and (vi) alkaloids. Many efforts have been undertaken to demonstrate the multidimensional effects of these compounds on overall health and well-being by influencing the immune system. The polysaccharide fraction, with emphasis on ITFs, of plant extracts used in herbal medicine has been largely overlooked in recent years. We speculate that these fractions may play an important role in the modulation of the immune system, either independently or synergistically. Further research efforts are required to identify and determine the functionality of these polysaccharides, especially ITFs, among various herbal medicines.

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
