# Peer review of "Fructans as Immunomodulatory and Antiviral Agents: The Case of Echinacea"

_biomolecules, 2019, doi:10.3390/biom9100615_

Round 1

Reviewer 1 Report

The review ‘Fructans as immunomodulatory and antiviral agents: the case of Echinacea’ by Erin Dobrange et al gives a broad and good overview on the bioactivities of a less explored compound class of oligosaccharides. First, the authors discuss the importance of herbal products, such as Echinacea preparations, in medicine. Then, some examples are given for immunomodulatory effects exerted by extracts of Echinacea, followed by an overview on the bioactivies of fructans with the conclusion that an overlap becomes evident between the effects observed for Echinacea and the fructans. This motivated the authors to look for fructans in Echinacea and investigate their immunomodulatory properties. I think that the discussion of fructans as bioactive ingredients in Echinacea extracts is interesting and merits publication; this review, however, would greatly benefit if the current literature on the actual occurrence of fructans in Echinacea were discussed in more detail. I believe that including a brief overview of phytochemical analyses of different plant parts in respect to fructans would give a more complete picture of the topic and attract more readers.

I think that the abstract and first chapter of the review, describing the importance of traditional medicine, are a bit misleading for the review; medicinal preparations based on plant extracts or isolated natural products from plants are not necessarily part of ‘traditional medicine’, but can also be the result of recent drug development efforts. I would recommend to rather use the wordings ‘herbal medicines’, ‘phytomedicines’, or 'phytopharmaceuticals’.

More importantly, I miss a discussion of the occurrence and identity of ITFs in Echinacea, since this is the central part of the review as stated in the title. The abstract states that ‘this review aims to highlight the occurrence of ITFs in Echinacea derived formulations’, but this topic is not addressed in the review. The one reference (Blaschke et al, reference 109) that is cited for the occurrence of ITFs in Echinacea is a study based on a root extract. Whether ITFs occur in e.g. pressed juices of aerial parts should be discussed.

The authors use their own results that were presented as a not peer-reviewed poster to demonstrate the occurrence of ITFs in commercially available Echinacea preparations in Belgium. A majority of the commercially available preparations of Echinacea are pressed juices of different parts of the plant. For the discussion of the results of Dobrange et al, 2019, it would be beneficial to mention which parts of the plant were used for the commercial preparations used for the study (which are indicated as P1-P5 on the poster with no further identification). I would further suggest a more detailed discussion of the currently available literature (or lack thereof) on analyses of fructans in Echinacea extracts. The composition of bioactive compounds in Echinacea extracts depends largely on the part of the plant used for extraction. Therefore, I think this review would profit substantially from a discussion of this subject.

Below are my more specific comments:

Page 2, lines 43-45: please change the definitions used for herbal products. Tinctures are ethanolic extracts, infusions and macerations are not necessarily hot and cold, and decoctions are made from boiled plant material (not tea).

Page 2, lines 62-62: ‘One of the most popular natural products are Echinacea preparations.’ The use of the expression ‘natural products’ for a preparation is confusing.  Compounds (i.e. small molecules) that are produced by a living organism are usually referred to as natural products.

Page 2, line 69: delete ‘of’

Page 2, line 76: Too generalized use of ‘Echinacea plant extracts’; the authors write that ‘studies demonstrated that Echinacea, as whole plant extract, could stimulate and signal immune responses’. The three references cited for this statement are two original articles and a review; in the case of reference 15, indeed the dried whole plant is used for the study, in reference 17, extracts of roots or stem+leaf are used, and reference 16 is a review on complementary and alternative medicines which states

‘Preparations are made from roots, above-ground parts (stems, leaves and flowers) or a mixture of both. Furthermore, some preparations of echinacea are aqueous extracts, whereas others are alcoholic extracts. Differences between the species, the soil in which the plants are cultivated, the parts of the plant used and the extraction procedures can result in substantial differences in chemical composition and biological activity. ... Immunological studies of echinacea have used aqueous or alcoholic extracts, as well as purified polysaccharides.’

Page 3, lines 86-87: ‘relatively little information is available about the polysaccharides present in Echinacea’: again, please discuss in more detail.

Page 3, line 91 and line 105: the abbreviation FOS is defined twice.

Page 3, line 93: What does ‘enhancing physiological features’ mean?

Page 3, lines 117-118: ‘respiratory infections (...) are often viral, infectious diseases caused by different viruses’; delete either ‘viral’ or ‘infectious diseases caused by different viruses.

Page 3, lines 118-119: The authors state that the only treatment against viral infections in modern medicine is vaccination. I disagree, there are plenty of antiviral drugs in clinical use.

Page 4, line 125: ‘a healthy homeostasis of the immune defense system (...) could play an important role in relieving the symptoms of viral infections’. Do the authors mean ‘in preventing viral infections’?

Page 4, line 156: this paragraph is part of the chapter dealing with anti-viral activities of fructans, however, discusses other effects of immunomodulation. Please either define a new chapter or move this paragraph to a different one.

Page 5, line 209: ‘The (...) properties of Echinacea preparations are well-known and has been described’ change to ‘have been’

Page 6, line 223: ‘another example of putative synergism’ change to ‘an example’, as this is the first time this topic appears in the review (no previous example).

Page 7, Line 265: ‘the present confirmation that the commercial Echinacea products contain these fructans’: which confirmation? No literature cited and not discussed before in the review. Which preparations? Only reference 109 is cited that investigates the occurrence of fructans in Echinacea (line 269). The further discussion of the occurrence of ITFs in commercially available Echinacea preparations is limited to a poster by the authors (Dobrange et al, 2019).

Page 7, line 289: ‘Since ITFs only represent a small part of these preparations...’ Please clarify, I understand that the abundance of ITFs in the tested preparations is low, and therefore the authors argue that prebiotic effects on their own cannot explain the observed immunomodulatory effects. Please rephrase.

Page 8, line 292: I would not call Indian Ayurveda and TCM ‘regional’

Page 8, line 294: ‘Currently, a holistic approach to the development of TMs is targeted for disease prevention.’ I do not understand this statement, please specify.

Page 8, lines 300-302: ‘Although many efforts have gone into demonstrating the multidimensional effects these compounds exert on overall health and well-being, of interest is the regulatory and adaptive aspects it has on the immune system.’ Please check the structure and correctness of the sentence.

Page 8, lines 302-307, final remarks: The review is focused on Echinacea, in which the occurrence of ITFs in not clarified in detail; the authors, however, wish for more research efforts to identify the function of ITFs in developing TMs. What are developing TMs? (As in chapter 1, the authors should consider using ‘phytomedicines’ instead) Do the authors wish for more phytochemical analyses of Echinacea extracts regarding ITFs?

Author Response

The review ‘Fructans as immunomodulatory and antiviral agents: the case of Echinacea’ by Erin Dobrange et al gives a broad and good overview on the bioactivities of a less explored compound class of oligosaccharides. First, the authors discuss the importance of herbal products, such as Echinacea preparations, in medicine. Then, some examples are given for immunomodulatory effects exerted by extracts of Echinacea, followed by an overview on the bioactivies of fructans with the conclusion that an overlap becomes evident between the effects observed for Echinacea and the fructans. This motivated the authors to look for fructans in Echinacea and investigate their immunomodulatory properties. I think that the discussion of fructans as bioactive ingredients in Echinacea extracts is interesting and merits publication;

>>>>We thank this reviewer for recognizing the unique angle in this exercise, since so far the putative immunomodulatory properties of fructans in Echinacea preparations have been largely neglected   

this review, however, would greatly benefit if the current literature on the actual occurrence of fructans in Echinacea were discussed in more detail. I believe that including a brief overview of phytochemical analyses of different plant parts in respect to fructans would give a more complete picture of the topic and attract more readers.

>>>>This is now taken into account in the revised version

I think that the abstract and first chapter of the review, describing the importance of traditional medicine, are a bit misleading for the review; medicinal preparations based on plant extracts or isolated natural products from plants are not necessarily part of ‘traditional medicine’, but can also be the result of recent drug development efforts. I would recommend to rather use the wordings ‘herbal medicines’, ‘phytomedicines’, or 'phytopharmaceuticals’.

>>>>We agree with this reviewer.  We now consistently used the wording 'herbal medicines' instead.   

More importantly, I miss a discussion of the occurrence and identity of ITFs in Echinacea, since this is the central part of the review as stated in the title. The abstract states that ‘this review aims to highlight the occurrence of ITFs in Echinacea derived formulations’, but this topic is not addressed in the review. The one reference (Blaschke et al, reference 109) that is cited for the occurrence of ITFs in Echinacea is a study based on a root extract. Whether ITFs occur in e.g. pressed juices of aerial parts should be discussed.

>>>ITFs indeed occur in different plant parts but the concentration and degree of polymerization ending up Echinacea products may greatly depend on a number of variables, including the organs from which they are extracted, the extraction method used (e.g. the percentage of ethanol used) and environmental as well as seasonal variations. Pressed juices of aerial parts or root extracts are both common. This is now better explained in the revised version.  

The authors use their own results that were presented as a not peer-reviewed poster to demonstrate the occurrence of ITFs in commercially available Echinacea preparations in Belgium. A majority of the commercially available preparations of Echinacea are pressed juices of different parts of the plant. For the discussion of the results of Dobrange et al, 2019, it would be beneficial to mention which parts of the plant were used for the commercial preparations used for the study (which are indicated as P1-P5 on the poster with no further identification). I would further suggest a more detailed discussion of the currently available literature (or lack thereof) on analyses of fructans in Echinacea extracts. The composition of bioactive compounds in Echinacea extracts depends largely on the part of the plant used for extraction. Therefore, I think this review would profit substantially from a discussion of this subject.

>>>>Details on P1-P5 plant part and extraction method used are available  through the poster weblink and this is now more thoroughly discussed in the revised version.

Below are my more specific comments:

Page 2, lines 43-45: please change the definitions used for herbal products. Tinctures are ethanolic extracts, infusions and macerations are not necessarily hot and cold, and decoctions are made from boiled plant material (not tea).

>>>Fixed in the revised version

Page 2, lines 62-62: ‘One of the most popular natural products are Echinacea preparations.’ The use of the expression ‘natural products’ for a preparation is confusing.  Compounds (i.e. small molecules) that are produced by a living organism are usually referred to as natural products.

>>>The term herbal products is used instead

Page 2, line 69: delete ‘of’

>>>Fixed

Page 2, line 76: Too generalized use of ‘Echinacea plant extracts’; the authors write that ‘studies demonstrated that Echinacea, as whole plant extract, could stimulate and signal immune responses’. The three references cited for this statement are two original articles and a review; in the case of reference 15, indeed the dried whole plant is used for the study, in reference 17, extracts of roots or stem+leaf are used, and reference 16 is a review on complementary and alternative medicines which states

‘Preparations are made from roots, above-ground parts (stems, leaves and flowers) or a mixture of both. Furthermore, some preparations of echinacea are aqueous extracts, whereas others are alcoholic extracts. Differences between the species, the soil in which the plants are cultivated, the parts of the plant used and the extraction procedures can result in substantial differences in chemical composition and biological activity. ... Immunological studies of echinacea have used aqueous or alcoholic extracts, as well as purified polysaccharides.’

>>>>This is now explained in the revised version

Page 3, lines 86-87: ‘relatively little information is available about the polysaccharides present in Echinacea’: again, please discuss in more detail.

>>>> This is more thoroughly discussed further on

Page 3, line 91 and line 105: the abbreviation FOS is defined twice.

>>>>Fixed

Page 3, line 93: What does ‘enhancing physiological features’ mean?

>>>Clarified in the revised version

Page 3, lines 117-118: ‘respiratory infections (...) are often viral, infectious diseases caused by different viruses’; delete either ‘viral’ or ‘infectious diseases caused by different viruses.

>>>The word viral has been deleted

Page 3, lines 118-119: The authors state that the only treatment against viral infections in modern medicine is vaccination. I disagree, there are plenty of antiviral drugs in clinical use.

>>>Indeed, the text has been amended accordingly

Page 4, line 125: ‘a healthy homeostasis of the immune defense system (...) could play an important role in relieving the symptoms of viral infections’. Do the authors mean ‘in preventing viral infections’?

>>>Yes, the text has been amended accordingly

Page 4, line 156: this paragraph is part of the chapter dealing with anti-viral activities of fructans, however, discusses other effects of immunomodulation. Please either define a new chapter or move this paragraph to a different one.

>>>>This paragraph was transferred to the beginning of Section 3, where it indeed fits better

Page 5, line 209: ‘The (...) properties of Echinacea preparations are well-known and has been described’ change to ‘have been’

>>>>Fixed

Page 6, line 223: ‘another example of putative synergism’ change to ‘an example’, as this is the first time this topic appears in the review (no previous example).

>>>Fixed

Page 7, Line 265: ‘the present confirmation that the commercial Echinacea products contain these fructans’: which confirmation? No literature cited and not discussed before in the review. Which preparations? Only reference 109 is cited that investigates the occurrence of fructans in Echinacea (line 269). The further discussion of the occurrence of ITFs in commercially available Echinacea preparations is limited to a poster by the authors (Dobrange et al, 2019).

>>>The confusion happened indeed because some statements appeared too late. We have added additional references earlier in the text and switched the sequence of announcing some sentences in such way that everything becomes more logic and straightforward, contributing to the overall flow.   

Page 7, line 289: ‘Since ITFs only represent a small part of these preparations...’ Please clarify, I understand that the abundance of ITFs in the tested preparations is low, and therefore the authors argue that prebiotic effects on their own cannot explain the observed immunomodulatory effects. Please rephrase.

>>>We have rephrased this sentence as requested

Page 8, line 292: I would not call Indian Ayurveda and TCM ‘regional’

>>>>Fixed

Page 8, line 294: ‘Currently, a holistic approach to the development of TMs is targeted for disease prevention.’ I do not understand this statement, please specify.

>>>This is now rephrased for clarification 

Page 8, lines 300-302: ‘Although many efforts have gone into demonstrating the multidimensional effects these compounds exert on overall health and well-being, of interest is the regulatory and adaptive aspects it has on the immune system.’ Please check the structure and correctness of the sentence.

>>>This sentence was rephrased now

Page 8, lines 302-307, final remarks: The review is focused on Echinacea, in which the occurrence of ITFs in not clarified in detail; the authors, however, wish for more research efforts to identify the function of ITFs in developing TMs. What are developing TMs? (As in chapter 1, the authors should consider using ‘phytomedicines’ instead) Do the authors wish for more phytochemical analyses of Echinacea extracts regarding ITFs?

>>>The term developing TMs was wrong and we deleted it. More research is required into the relative importance of ITFs in ITF containing Echinacea products.

We are really very thankful to this reviewer for providing so many good suggestions that allowed us to further increase the quality, consistency and overall structure of this review 

Reviewer 2 Report

While plants have been long studied for sources of potential therapeutics, the field of drug discovery for immunomodulatory compounds is still relatively new, and there exists a great need for development of new drug classes.  This review outlines the role of Echinacea as a traditional medicine, and in particular, highlights inulin-type fructans from these spp.  Overall, this is an interesting paper.  There have been a lot of recent reviews on inulin, but not echinacea-derived inulin.  The particular emphasis on synergistic effects with other compounds in echinacea is novel, but that section could be expanded more to differentiate this review from the plethora of other inulin reviews of 2019.

Minor concerns:

1 - Echniacea is well known as a TM with immunomodulatory effects.  What isn't clear is why the review focuses on ITFs.  The justification for the review needs to be set up better.  For example, why is there the supposition that ITFs are more important than the more well known polyphenol and alkylamide fractions of echinacea?

2 - In general, the manuscript jumps from topic to topic without a clear flow.  For example, why (from 2-3) does the review go from discussing echinacea to intestinal flora?  It takes a few readings to understand why the authors shift topics, which makes the review less clear.  The review needs to be restructured better, or at least have additional linking paragraphs to improve flow.

3 - section 3.1 discusses antibiotic treatment against viral infections.  Of course there is not, but there are anti-viral treatments.  The only MM treatment is not vaccination.

4 - In general, there are too many acronyms.  CC is used once or twice, when it would just be clearer to the reader to write out common cold.  Unless acronyms are used a lot and on nearly every page, or are common (like TLRs), then they should just write out the word.

5 - The paragraph starting on line 269 is awkward.  Why do they highlight their own research, especially with a different reference style and doi?  This work should be put into context of the subject of the review, as all others also are.

6 - Why is there a focus on echinacea in the Belgian market only?  The readers of biomolecules are international and so this information and section of the paper should be more global.

Author Response

While plants have been long studied for sources of potential therapeutics, the field of drug discovery for immunomodulatory compounds is still relatively new, and there exists a great need for development of new drug classes.  This review outlines the role of Echinacea as a traditional medicine, and in particular, highlights inulin-type fructans from these spp.  Overall, this is an interesting paper.  There have been a lot of recent reviews on inulin, but not echinacea-derived inulin.  The particular emphasis on synergistic effects with other compounds in echinacea is novel, but that section could be expanded more to differentiate this review from the plethora of other inulin reviews of 2019.

>>>We are happy that this reviewer recognizes the novel aspects tat are highlighted in this review. We extended and re-organized the text in such way to make it even more specific (as compared to other inulin reviews) and more straightforward.  

Minor concerns:

1 - Echniacea is well known as a TM with immunomodulatory effects.  What isn't clear is why the review focuses on ITFs.  The justification for the review needs to be set up better.  For example, why is there the supposition that ITFs are more important than the more well known polyphenol and alkylamide fractions of echinacea?

>>>We did not make any statements that the relative importance of ITFs would be higher than those of alkamide and polyphenol fractions. That is a matter of deeper investigation. Fact is that the scientific community so far did not pay a lot of attention to the presence of ITFs in Echinaceae preparations and this in itself is enough justification for this review. This is now better introduced and build-up in the revised version.     

2 - In general, the manuscript jumps from topic to topic without a clear flow.  For example, why (from 2-3) does the review go from discussing echinacea to intestinal flora?  It takes a few readings to understand why the authors shift topics, which makes the review less clear.  The review needs to be restructured better, or at least have additional linking paragraphs to improve flow.

>>>The structure of the manuscript has been optimized. It is essential to highlight a number of processes happening in the gut system since this is the exact spot where direct and indirect effects of ITFs (and SCFA derived thereof) are initiated, with important links to human health and well being as well as to brain functioning (see Dalile et al., 2019; Nature Reviews 16: 461-474). 

3 - section 3.1 discusses antibiotic treatment against viral infections.  Of course there is not, but there are anti-viral treatments.  The only MM treatment is not vaccination.

>>>The text has been amended

4 - In general, there are too many acronyms.  CC is used once or twice, when it would just be clearer to the reader to write out common cold.  Unless acronyms are used a lot and on nearly every page, or are common (like TLRs), then they should just write out the word.

>>>The number of acronyms was decreased

5 - The paragraph starting on line 269 is awkward.  Why do they highlight their own research, especially with a different reference style and doi?  This work should be put into context of the subject of the review, as all others also are.

>>>>Fixed

6 - Why is there a focus on echinacea in the Belgian market only?  The readers of biomolecules are international and so this information and section of the paper should be more global.

>>>Actually these products are widespread over Europe and beyond.